# Differences in Cognitive Function in Women and Men with Diabetic Peripheral Neuropathy with or without Pain

**DOI:** 10.3390/ijerph192417102

**Published:** 2022-12-19

**Authors:** Jenifer Palomo-Osuna, Inmaculada Failde, Helena De Sola, María Dueñas

**Affiliations:** 1The Observatory of Pain, University of Cádiz, 11009 Cádiz, Spain; 2Research Unit, Biomedical Research and Innovation Institute of Cádiz (INiBICA), Puerta del Mar University Hospital, University of Cadiz, 11009 Cádiz, Spain; 3Preventive Medicine and Public Health Area, University of Cádiz, 11009 Cádiz, Spain; 4Department of Statistics and Operational Research, University of Cádiz, 11510 Cádiz, Spain

**Keywords:** painful diabetic neuropathy, cognitive function, sex, gender

## Abstract

The aim of this study was to analyse the differences in cognitive function between women and men with type-2 diabetes mellitus (DMT2) and diabetic peripheral neuropathy (DPN) with and without diabetic neuropathic pain (DNP), and the factors associated with cognitive function in each sex. A cross-sectional study of 149 patients with DMT2 and DPN was performed. Sociodemographic and clinical variables, Test Your Memory (TYM) for cognitive assessment, anxiety and depression (HADS), quality of life (SF-12v2) and sleep characteristics (MOS-sleep) were measured. A high percentage of women presented cognitive impairment (50% vs. 36.1%) and they scored lower on the TYM (mean = 40.77; SD = 6.03 vs. mean = 42.49; SD = 6.05). Women with DNP scored lower on calculation tasks (3.17 vs. 3.52) than men with DNP, while women without DNP scored lower on retrograde memory (2.70 vs. 3.74), executive function (3.83 vs. 4.25) and similarities (2.51 vs. 3.12) than men without DNP. Being older (B = −0.181) and presenting cardiovascular risk factors (B = −5.059) were associated with worse cognitive function in women, while in men this was associated with older age (B = −0.154), a longer duration of diabetes (B = −0.319) and the presence of depression (B = −0.363). Women with and without DNP obtained worse results in cognitive function. However, the presence of pain had a greater impact on the different dimensions in men.

## 1. Introduction

In recent years, differences have been reported between women and men affected by many diseases with studies showing that sex and gender can be important risk markers [1]. Furthermore, several authors report that both genders, related to social constructs, and sex, referring to biological determinants [2], are relevant for understanding the differences in clinical manifestations and care outcomes in women and men [2,3].

Type 2 diabetes mellitus (DMT2) is an important chronic disease where these differences have been revealed [3]. In this vein, differences in the prevalence of DMT2 at different stages of life have been observed between men and women. The disease has been reported to be more prevalent among women during puberty and adolescence, in men in middle age, and presents a similar frequency among old people [4]. Different factors have been associated with these results, such as biological factors related with hormone, phenotype and insulin resistance differences, and social and healthcare factors associated with longer delays in diagnosing the disease and the worse metabolic control described among the female population [4,5,6]. Likewise, men with DMT2 report having a better physical quality of life than women [7].

Diabetic peripheral neuropathy (DPN) is a very common complication of DMT2; pain being a problem that frequently affects these patients, several authors having shown differences between men and women [8,9]. Specifically, Samulowitz et al. [10] observed that pregnancy, menstrual cycle and oral contraceptive use can mediate women’s response to pain. Moreover, these authors have stated that men and women behave differently when dealing with chronic pain (CP), showing that, in general, men are more stoic, tolerating and denying pain and avoiding seeking health care, while women are more sensitive and more willing to report pain, with it generally being more acceptable for women to express their feelings and talk about it [10].

Regarding cognitive function, several authors [11,12] have found greater cognitive impairment in women, explaining that this is associated with their greater longevity or the differences in dealing with cognitive domains whereby men have better visuospatial skills, while women deal better with verbal tasks in which they feel more self-efficacy. Authors such as Petersen [11] and Elosúa et al. [12] relate the different skills of men and women with gender stereotypes, such as the fact that boys tend to play with building toys that involve handling and transforming objects, while girls prefer reading and writing activities, habits that are encouraged by teachers at school.

Despite the differences observed between men and women with DMT2 and in subjects with cognitive impairment, and the evidence of cognitive impairment in patients with DMT2 and diabetic neuropathy [3,12,13,14,15,16], to our knowledge, the difference in cognitive function between men and women with DMT2 and DPN with or without diabetic neuropathic pain (DNP) has not been analysed. In addition, taking into account the higher prevalence in women of anxiety, depression and sleep disorders [17,18,19,20], which are very often related to cognitive impairment and DPN [13,20,21,22], it could be of great interest to analyse these relationships in patients with DMT2 and DPN and neuropathic pain.

Based on all the above, this study aimed to analyse the differences that exist in sociodemographic and clinical variables between women and men with DMT2 and DPN, and to analyse the difference in cognitive function between women and men with DMT2 and DPN with and without DNP.

## 2. Materials and Methods

### 2.1. Participants

A multi-centre cross-sectional study was conducted between July 2017 and March 2018. Participants were >18 years old and presented with diabetes mellitus type 2 (DMT2), diagnosed using the American Diabetes Association (ADA) criteria and included in the “*Diabetes Mellitus Integral Care Process*” (PAIDM) in six primary care centres in Cádiz (Andalusia, Spain). The DMT2 patients were selected if they had diabetic neuropathy diagnosed by a clinical examination and the monofilament test used in clinical foot examinations.

The exclusion criteria were patients that did not meet the criteria for inclusion, those who had some neurodegenerative disease, such as frontotemporal dementia or Alzheimer, or a mental illness or physical disease that prevented completion of the scales used in this study.

### 2.2. Selection Process

The participants were selected using non-probabilistic consecutive sampling of high-risk patients who met at least one of the following four indicators: presence of diabetic retinopathy, existence of foot ulcers, *HbA_1c_* > 8% in the last year, or a diagnosis of diabetes over 10 years previously. To confirm the existence of diabetic neuropathy, well-trained researchers performed the monofilament test on the patients. This procedure involves examining 10 reference points on each foot, 8 plantar and 2 dorsal. Diabetic neuropathy was confirmed if the patient did not detect the monofilament in three or more points on one foot [23].

The researchers contacted the selected patients by telephone to explain the study and invite them to participate. The patients that agreed to participate were cited in the corresponding healthcare centre, informed about the study objectives and invited to sign the informed consent.

A total of 149 patients with DMT2 and diabetic neuropathy were included. The selected patients were classified into two groups with and without DNP, according to the result of the “Douleur Neuropathique 4 questionnaire (DN4)” [24]. This scale consists of 10 binary items providing a final score from 0 to 10 with the subjects scoring 4 or more identified as presenting DNP. This scale has been adapted and validated in Spanish and has a sensitivity of 79.8% and a specificity of 78.0% [24].

### 2.3. Instruments and Variables

Information about sociodemographic data (sex, age, education and employment status) and clinical information (duration of DPN, treatments, DMT2 complications and cardiovascular risk factors) were collected from the patients using a clinical interview.

The Test Your Memory screening test (TYM) was used to evaluate cognitive function. This tool consists of 10 items with an overall score ranging from 0 to 50 calculated through 10 cognitive dimensions: executive, anterograde memory, visuospatial ability, naming, similarities, verbal fluency, calculation, retrograde memory, copying and orientation. The cut-off point is 42/50 (≤41 points indicate cognitive dysfunction) and a higher score indicates better cognitive performance [25,26]. This scale was adapted and validated for chronic pain patients by our research group [27,28].

The visual analogue scale (VAS) was used to evaluate the pain intensity. The score ranges from 0 to 10, with 0 referring to no pain and the highest score the worst pain possible.

The neuropathic pain symptom inventory (NPSI) was used to measure the neuropathic pain phenotype. This scale includes ten items that allow five dimensions to be calculated: evoked pain, deep spontaneous pain, superficial spontaneous pain, paroxysmal pain and paraesthesia/dysesthesia. The score for each dimension ranges from 0 to 10, with 0 being the absence of this phenotype and the highest score being the maximum presence. This scale was also validated and adapted in Spanish [29].

The Hospital Anxiety and Depression Scale (HADS) was used to measure anxiety and/or depression. This scale has two subscales: HADS-A (anxiety) and HADS-D (depression). Each subscale has seven items and a 21-point maximum score and >10 points suggest the presence of anxiety and/or depression. This scale was validated in a Spanish population [30,31] and in patients with DNP [32].

The Medical Outcomes Sleep (MOS) scale was used to evaluate sleep quality. It includes a summary index for measuring sleep quality (Index-9) and 12 items that explore the effect of the illness on sleep dimensions. The score ranges from 0 to 100 and a higher score shows more sleep problems. This scale was validated in a Spanish population and is very useful for evaluating sleep problems in patients with DNP [33].

The SF-12v2 health survey was used to measure health-related quality of life (HRQL) [34]. It contains 12 items and eight dimensions calculated through the items: physical functioning (PF), role-physical (RP), bodily pain (BP), general health perception (GH), vitality (V), social functioning (SF), role-emotional (RE) and mental health (MH). Moreover, this instrument has two global dimensions, the physical health component summary (PSC-12) and mental health component summary (MSC-12), scoring from 0 to 100 with the highest scores indicating a better quality of life.

### 2.4. Statistical Analysis

A descriptive analysis was performed in both groups of patients, women and men. In the case of the qualitative variables, absolute (n) and relative (%) frequencies were used, while for the quantitative variables, measures of centralisation (mean) and dispersion (standard deviation (SD)) were calculated to compare the characteristics of the patients and the differences in cognitive function between women and men. The chi-square test, *t* tests, ANOVA, Mann–Whitney U, Kruskal–Wallis and correlation coefficients (Pearson or Spearman) were used, according to the type and distribution of the variables, previously evaluated by the Kolmogorov–Smirnoff test. A *p*-value of < 0.05 was considered statistically significant.

Furthermore, two multiple linear regression models were performed, one for women and another for men, in which the dependent variable was the total TYM score and the independent variables included in the models were depression, anxiety, sleep, HRQL and the sociodemographic and clinical variables described above. The stepwise method was used to select the variables of the model and R^2^ was considered as the goodness-of-fit measure. The analyses were performed with the IBM SPSS v.24 statistical package.

## 3. Results

### 3.1. Characteristics of Women and Men with DPN

Among the 149 subjects with DPN included in this study, 66 were women and 83 were men. The women were older (72.91 vs. 70.45), more frequently had no formal education (39.4% vs. 30.1%) and more of them were homemakers than men (68.2% vs. 1.2%, *p* < 0.001) (Table 1). On the contrary, the majority of the men were retired (66.3% vs. 15.2%).

Differences were not found between the groups in any clinical variables related to DMT2, or in complications or cardiovascular risk factors except for obesity, which was more frequent in women (53% vs. 30.1%) (Table 1). Of note is that more women took medication that affected cognition (74.2% vs. 47%), pills for sleeping (55.6% vs. 31.6%) and painkillers (71.4% vs. 50.6%) (Table 1).

Regarding the scales, the scores on HADS anxiety (7.81 vs. 4.66) and mental (7.56 vs. 5.09) and Index-9 (41.06 vs. 33.04) were higher in women, meaning that they have more anxiety, depression and sleep problems. Moreover, women had lower scores on the mental summary components of the SF-12 (44.82 vs. 51.28), indicating worse mental health (Table 1).

The result of the TYM showed a higher percentage of women than men with a cognitive score ≤ 41 (cognitive dysfunction) (50% vs. 36.1%), although the result was not statistically significant. Likewise, the mean scores on this scale were lower in women than in men (40.77 vs. 42.49, *p* = 0.048) (Table 2). Regarding the dimensions of the TYM, lower scores were observed in women in retrograde memory (2.06 vs. 2.36), calculation (3.37 vs. 3.57), similarities (2.62 vs. 3.13) and executive function (3.84 vs. 4.22) (Table 2).

Concerning the presence of diabetic neuropathic pain (DNP), 35 women and 36 men were with DNP, and 32 women and 47 men were without DNP.

### 3.2. Differences in TYM Results in Women and Men with and without DNP

Regarding the results by sex in patients with and without DNP, we observed a higher percentage of women showing a TYM score ≤ 41 in both groups, with DNP (45.7% vs. 36.1%) and without DNP (54.8% vs. 36.2%) (Table 3). Similar results were observed in the mean TYM scores, where the mean score in women was lower in both groups, with DNP (40.68 vs. 42.22) and without DNP (40.87 vs. 42.70) (Table 3).

Concerning the dimensions of the TYM, the only difference observed between women and men with DNP was in calculation (3.17 vs. 3.52) (Table 3). However, in the subjects without DNP, we found differences in the dimensions related to similarities (2.51 vs. 3.12), anterograde memory (2.70 vs. 3.74) and executive function (3.83 vs. 4.25); the women’s scores were always lower than those of the men (Table 3).

### 3.3. Factors Related to Cognitive Function (Test Your Memory) in Women and Men with DPN, Multiple Linear Regression Models

The analysis of the variables associated with the TYM score in women showed that the older women (B = −0.181) and those with cardiovascular risk factors (B = −5.059) presented lower TYM scores (worse cognitive function). On the contrary, a higher educational level was related to a higher score on the TYM (better cognitive function). Taking medicine that affected cognition was included as an adjustment variable in this model (Table 4).

Regarding the variables related to the TYM score in the men, we observed that those with a longer diabetes duration (B = −0.319) and high scores for depression on the HAD scale (B = −0.389) achieved lower TYM scores. On the contrary, a higher educational level, the presence of AHT (B = 2.549) and having higher scores on the HAD anxiety scale (B = 0.551) and on the mental component of the SF-12v2 (B = 0.181) were related to higher scores on the TYM. Taking medication that affected cognitive function and age were included as adjustment variables in this model (Table 4).

## 4. Discussion

This study analyses the differences in cognitive function and in specific dimensions of cognitive performance between women and men with DMT2 and DPN and with and without DNP, as well as the variables related to cognitive function in each sex.

The results highlight that the women obtained lower mean scores on the TYM scale, which is an indication of greater cognitive impairment and a higher prevalence of cognitive impairment than the men, both in the group of patients with DNP and in the group without.

Likewise, it is worth underlining that when comparing the men and women with pain, differences are only observed in the dimension of cognitive function that explores calculation. However, differences were found in nearly all the dimensions when comparisons were performed by sex among the patients without pain. These results could suggest that pain has more serious consequences in men than in women, who also present cognitive impairment in the absence of pain.

The risk of women suffering from cognitive impairment has been reported in the general population [35] and in patients with DMT2 [1,36], and has been related to a lower educational level or the type of occupation, such as housework, traditionally done by a higher percentage of women and not requiring much intellectual effort, which has been shown to improve cognitive performance in adults [35]. Likewise, other studies have explained that these differences are the result of a less healthy lifestyle, such as a sedentary life or an inappropriate diet [37] or the higher prevalence of depression among women [18].

Concerning the differences found in specific dimensions of cognitive function in the population without pain, our results are in line with those of other studies that show that women obtain lower scores in dimensions such as calculation or anterograde memory, also explained by a lower education level, which leads to reduced access to intellectual and cultural activities and to complex tasks that encourage the development of a good cognitive reserve in adulthood [35].

A particularly novel finding in this study is the greater effect of pain observed on the cognitive function of the men with DNP. Previous studies have shown that both acute and chronic pain are associated with reduced cognitive performance [14,38] with this effect not only being evident on a general level, but also in specific areas such as attention, memory, processing speed and executive function [38]. Likewise, some authors have found a risk of cognitive impairment in women with chronic pain [38]. However, to our knowledge, no studies have been published that analyse cognitive function in men and women both with and without DNP or address the consequences that pain may have in men and women when a cognitive deficit already exists, as we observed in our study, where the scores on the different dimensions of the TYM were the same for the men and women suffering from pain.

It is worth highlighting that among the subjects with DNP only calculation was more affected in the women than in the men. Gender stereotypes hold that males outperform females in mathematics and spatial tests, while the opposite is true in verbal tests, which has been related to the gender stratification hypothesis, which maintains that gender differences in outcomes such as maths performance are closely related to opportunity structures for girls and women in their culture [39,40]. However, a meta-analysis performed by Hyde [41] found that, while culture-dependent, these differences have disappeared in recent years, indicating that females have reached parity with males in maths performance today. This would not seem to be supported by our results, although it could be explained by the age of the study population, who had a mean age of over 70.

Regarding the results of the factors associated with cognitive function, we observed similarities in both men and women in that being older and having a lower educational level was associated with worse cognitive function scores. Age is known to be the greatest risk factor for cognitive impairment and Alzheimer’s disease [5]. In the same vein, educational level has frequently been shown to affect cognitive function [42].

Regarding the differences in associated factors according to gender, it was observed that impaired cognitive function in the women was associated with the presence of cardiovascular risk factors, while among the men it was related to increased depression and the duration of the diabetes. Surprisingly, anxiety and AHT had a positive effect on cognitive function in the men.

In agreement with the results found for the women in this study, other authors have reported a greater frequency of cardiovascular risk factors such as abdominal obesity, insulin resistance and dyslipidaemia, all of which are related to impaired cognitive function in the female population [5,43].

Regarding the duration of the diabetes, in other studies the evolution time of DMT2 has been inversely related [36,37] with cognitive function and with a higher risk of suffering from DPN [37,42], and has been attributed to the greater likelihood of suffering from swings in blood sugar levels and higher levels of *HbA_1c_*. This may be particularly relevant in men, who present a greater prevalence of DMT2 in middle age than women [4,5] although we have not been able to demonstrate it in this paper as it was a cross-sectional study in which the levels of *HbA_1c_* found were similar in the men and women.

Regarding depression, authors such as Almeida et al. [44] found that prodromal symptoms of depression are associated with the later onset of cognitive impairment in men. Likewise, Palomo-Osuna et al. [13] found that depression is also related with the greater presence of cognitive impairment in people with DPN, although the results of the systematic review are not differentiated by sex.

The relationship between the presence of anxiety and cognitive impairment observed in the men in this study could be explained by the processing efficiency theory, which reveals that the worrying that is part of the state of anxiety can lead to maximum performance of a task when anxiety is moderate due to the combination of processes affecting the working memory and the increased effort by the individual when performing the task [45]. Likewise, fluctuations in anxiety may also have influenced our results [45]. However, neither the study design nor the instrument used to assess anxiety (HADS) enabled these fluctuations to be determined, so we cannot confirm this theory.

Finally, regarding the relationship between AHT and cognitive function in men, although the literature generally associates AHT with worse cognitive function, some studies report the opposite [46]. In this sense, authors such as Heijer et al. [47] reflect that since the central nervous system is involved in controlling blood pressure, it is reasonable to consider that in subjects presenting brain atrophy, this could lead to a decrease in blood pressure. However, it is unlikely that the patients in this study were suffering from significant atrophy since those with Alzheimer’s disease or cognition problems that prevented them from completing the scales were excluded. In addition, they did not undergo MRI testing that could have confirmed it.

As a strength of this study, we can highlight the presentation of novel results that have never been analysed in the population of patients with DMT2 and DPN, where pain seems to affect men and women differently and where the factors associated with impaired cognitive function in men and women suffering from neuropathic pain are different. Likewise, the use of validated scales enables high-quality information to be obtained about the study population.

Moreover, the multi-centre design of this study is of note as it made it possible to obtain a larger sample. However, it is important to note that it was not possible to reach the sample size calculated initially since we had to stop the data collection ahead of schedule due to the COVID-19 pandemic.

A limitation of this study is its cross-sectional design, which does not allow for causal relationships to be established between the variables studied. Likewise, variables such as ethnicity and physical activity, which could have provided interesting results, were not included. Finally, the instrument used to measure anxiety and depression (HADS) uses the past-week time frame and not the moment in which cognitive function was assessed, possibly leading to an offset in the assessment. However, this instrument is widely used in studies of both pain and cognitive function.

## 5. Conclusions

This study shows that women with DMT2 and DPN with and without DNP present worse cognitive function than men, and that the effect of pain on the different cognitive dimensions differs between men and women. Likewise, this study presents differences in the variables that affect cognitive functioning in both groups, cardiovascular risk being more important among the women, while a longer duration of the DMT2 and the presence of depression were the most significant variables among the men.

Given that over many years both basic research studies and clinical studies were conducted only with male subjects, this study shows the need to perform epidemiological studies that analyse the two sexes separately. In such a way, risk factors could be identified that may be different, affecting the possible responses to the treatments. Likewise, within clinical practice, identifying these factors may facilitate a therapeutic intervention the achieves better results in each sex.

## Figures and Tables

**Table 1 ijerph-19-17102-t001:** Characteristics of women and men with DPN.

Variables N = 149	Women N = 66	Men N = 83	*p*
Sociodemographic data			
**Age**	**Mean (Standard Deviation)** 72.91 (8.45)	**Mean (Standard Deviation)** 70.45 (9.55)	0.125 ^a^
	**N (%)**	**N (%)**	
**Education level** No formal education Primary studies Secondary and University studies	26 (39.4) 29 (43.9) 11 (16.7)	25 (30.1) 39 (47) 19 (22.9)	0.426 ^b^
**Employment status** Unemployed Homemaker Working Retired Partial and total disability	4 (6.1) 45 (68.2) 2 (3.0) 10 (15.2) 5 (7.6)	4 (4.8) 1 (1.2) 2 (2.4) 55 (66.3) 21 (25.3)	<0.001 ^c^
**Clinical data**			
**Time since type-2 diabetes mellitus diagnosis (years) N= 140**	**Mean (Standard Deviation)** 11.86 (3.34)	**Mean (Standard Deviation)** 11.37(3.60)	0.785 ^a^
** *HbA_1c_* ** **registered N = 146**	**Mean (Standard Deviation)** 7.50 (1.50)	**Mean (Standard Deviation)** 7.53 (1.46)	0.785 ^a^
	**N (%)**	**N (%)**	
**Medication that affects cognition** Yes	49 (74.2)	39 (47)	0.001 ^b^
**Medication for sleep** Yes	35 (55.6)	25 (31.6)	0.004 ^b^
**Medication for pain** Yes	45 (71.4)	40 (50.6)	0.012 ^b^
**Treatment with insulin** Yes	31 (47)	46 (55.4)	0.305 ^b^
**Physical comorbidity** Yes	60 (90.9)	67 (80.7)	0.082 ^b^
**Previous history of anxiety** Yes	19 (28.8)	13 (15.7)	0.053 ^b^
**Previous history of depression** Yes	19 (28.8)	13 (15.7)	0.053 ^b^
**Associated complications**			
	**N (%)**	**N (%)**	
**Complications** Yes	42 (63.6)	60 (72.3)	0.586 ^b^
**Complications’ number** 0 1 2 3 4	24 (36.4) 23 (34.8) 15 (22.7) 4 (6.1) 0 (0.0)	23 (27.7) 31 (37.3) 20 (24.1) 7 (8.4) 2 (2.4)	0.465 ^c^
**Diabetic retinopathy** Yes	14 (21.2)	24 (28.9)	0.284 ^b^
**Diabetic nephropathy** Yes	15 (22.7)	16 (19.3)	0.606 ^b^
**Diabetic foot** Yes	10 (15.2)	21 (25.3)	0.129 ^b^
**Cardiovascular disease** Yes	23 (34.8)	39 (47)	0.135 ^b^
**Cardiovascular risk factors**			
**Obesity** Yes	35 (53)	25 (30.1)	0.005 ^b^
**Arterial hypertension** Yes	50 (76.9)	56 (67.5)	0.206 ^b^
**Dyslipidaemia N = 70** Yes	41 (63.1)	47 (56.6)	0.428 ^b^
**Scales**			
	**Mean (Standard Deviation)**	**Mean (Standard Deviation)**	
**HADS Anxiety total**	7.81 (5.42)	4.66 (4.79)	<0.001 ^a^
**HADS Depression score**	7.56 (5.56)	5.09 (4.28)	0.007 ^a^
**Index 9**	41.06 (24.32)	33.04 (21.17)	0.058 ^a^
**US Standardised physical component**	36.25 (12.79)	39.40 (12.53)	0.133 ^d^
**US Standardised mental component**	44.82 (14.31)	51.28 (13.12)	0.004 ^a^

^a^ Mann–Whitney; ^b^ Chi-square test; ^c^ Likelihood ratio; ^d^ t-Student.

**Table 2 ijerph-19-17102-t002:** Differences in TYM domains in women and men with DPN.

TYM Scale: Total Score and Domains			
TYM total Score	40.77 (6.03)	42.49 (6.05)	0.048 ^a^
TYM categorical With deterioration Without deterioration	33 (50%) 33 (50%)	30 (36.1%) 53 (63.9%)	0.089 ^b^
Orientation (place & person orientation)	9.81 (0.46)	9.56 (0.95)	0.061 ^a^
Copying	1.21 (0.88)	1.39 (0.83)	0.192 ^a^
Retrograde memory (semantic knowledge)	2.06 (0.96)	2.36 (0.82)	0.049 ^a^
Calculation	3.37(0.85)	3.57 (0.88)	0.018 ^a^
Verbal fluency	3.71 (0.84)	3.66 (0.85)	0.472 ^a^
Similarities	2.62 (1.21)	3.13 (1.13)	0.005 ^a^
Naming	4.87 (0.57)	4.83 (0.58)	0.472 ^a^
Visuospatial ability 1 & 2 (clock)	6.10(1.26)	6.08 (1.45)	0.596 ^a^
Anterograde memory	3.13 (1.96)	3.65 (1.96)	0.099 ^a^
Executive (ability to complete the test)	3.84 (0.79)	4.22 (0.86)	0.002 ^a^

^a^ Mann–Whitney; ^b^ Chi-square test.

**Table 3 ijerph-19-17102-t003:** Differences in the TYM results in women and men with and without DNP.

Variables N = 149	Women DNP N = 35	Men DNP N = 36	Women without DNP N = 31	Men without DNP N = 47	P1	P2	P3	P4
**TYM Total Score and Domains**								
**TYM total Score**	**M (SD)** 40.68 (6.72)	**M (SD)** 42.22 (6.67)	**M (SD)** 40.87 (5.25)	**M (SD)** 42.70 (5.60)	0.256 ^a^	0.103 ^a^	0.902 ^d^	0.879 ^a^
**TYM categorical** **With cognitive impairment** **Without cognitive impairment**	16 (45.7%) 19 (54.3%)	13 (36.1%) 23 (63.9%)	17 (54.8%) 14 (45.2%)	17 (36.2%) 30 (63.8%)	0.411 ^b^	0.104 ^b^	0.459 ^b^	0.996 ^b^
**Orientation (place & person orientation)**	**M (SD)** 9.85 (0.42)	**M (SD)** 9.52 (1.08)	**M (SD)** 9.77 (0.50)	**M (SD)** 9.59 (0.85)	0.053 ^a^	0.470 ^a^	0.386 ^a^	0.694 ^a^
**Copying**	**M (SD)** 1.22 (0.87)	**M (SD)** 1.30 (8.88)	**M (SD)** 1.19 (0.90)	**M (SD)** 1.46 (0.77)	0.668 ^a^	0.187 ^a^	0.899 ^a^	0.449 ^a^
**Retrograde memory (semantic knowledge)**	**M (SD)** 1.97 (1.01)	**M (SD)** 2.33 (0.89)	**M (SD)** 2.16 (0.89)	**M (SD)** 2.38 (0.77)	0.103 ^a^	0.292 ^a^	0.483 ^a^	0.992 ^a^
**Calculation**	**M (SD)** 3.17 (0.98)	**M (SD)** 3.52 (0.94)	**M (SD)** 3.61 (0.61)	**M (SD)** 3.61 (0.85)	0.021 ^a^	0.482 ^a^	0.039 ^a^	0.782 ^a^
**Verbal fluency**	**M (SD)** 3.54 (1.01)	**M (SD)** 3.52 (0.97)	**M (SD)** 3.90 (0.54)	**M (SD)** 3.76 (0.73)	0.712 ^a^	0.246 ^a^	0.042 ^a^	0.102 ^a^
**Similarities**	**M (SD)** 2.71 (1.12)	**M (SD)** 3.13 (1.09)	**M (SD)** 2.51 (1.31)	**M (SD)** 3.12 (1.17)	0.075 ^a^	0.030 ^a^	0.605 ^a^	0.872 ^a^
**Naming**	**M (SD)** 4.77 (0.77)	**M (SD)** 4.83 (0.60)	**M (SD)** 5 (0.00)	**M (SD)** 4.82 (0.56)	0.673 ^a^	0.062 ^a^	0.054 ^a^	0.753 ^a^
**Visuospatial ability 1 & 2 (clock)**	**M (SD)** 6.05 (1.39)	**M (SD)** 6.30 (1.19)	**M (SD)** 6.16 (1.12)	**M (SD)** 5.91 (1.61)	0.388 ^a^	0.977 ^a^	0.989 ^a^	0.397 ^a^
**Anterograde memory**	**M (SD)** 3.51 (1.83)	**M (SD)** 3.52 (2.18)	**M (SD)** 2.70 (2.03)	**M (SD)** 3.74 (1.80)	0.713 ^a^	0.028 ^a^	0.098 ^a^	0.819 ^a^
**Executive (ability to complete the test)**	**M (SD)** 3.85 (0.91)	**M (SD)** 4.19 (0.98)	**M (SD)** 3.83 (0.63)	**M (SD)** 4.25 (0.77)	0.088 ^a^	0.008 ^a^	0.088 ^a^	0.909 ^a^

M = Mean; SD = Standard deviation; P1 Women with DNP vs. Men with DNP; P2 Women without DNP vs. Men without DNP; P3 Women with DNP vs. Women without DNP; P4 Men with DNP vs. Men without DNP; ^a^ Mann–Whitney; ^b^ Chi-square test; ^d^ t-Student.

**Table 4 ijerph-19-17102-t004:** Factors related to cognitive function (Test Your Memory) in women and men with DPN, multiple linear regression models.

Model 1 TYM Score Women: Multiple Linear Regression Model.					
Variable	Category/Unit	B (SE)	95% CI		*p*
Constant		55.188 (7.068)	41.039	69.337	<0.001
Age		−0.181 (0.083)	−0.346	−0.016	0.032
Education level (Ref. without formal instruction)	Primary studies	3.839 (1.521)	0.794	6.884	0.014
	Secondary and University studies	5.390 (2.024)	1.340	9.441	0.010
Cardiovascular risk factors	Yes No *	−5.059 (2.551)	−10.165	0.48	0.052
Medication for cognition	Yes No *	1.198 (1.626)	−2.058	4.453	0.465
**Model 2 TYM score men: Multiple linear regression model.**					
**Variable**	**Category/Unit**	**B (SE)**	**95% CI**		** *p* **
Constant		44.385 (7.410)	29.595	59.175	<0.001
Age		−0.154 (0.070)	−0.293	−0.015	0.031
Education level (Ref. without formal education)	Primary studies	2.727 (1.373)	−0.014	5.467	0.051
	Secondary and University studies	5.400 (1.745)	1.918	8.883	0.003
Time since type-2 diabetes mellitus diagnosis (years)		−0.316 (0.167)	−0.649	0.016	0.016
HADS Anxiety score		0.506 (0.190)	0.127	0.886	0.010
HADS Depression score		−0.363 (0.192)	−0.747	0.020	0.063
US Standardised mental component		0.163 (0.071)	0.021	0.306	0.026
AHT		2.549 (1.277)	0.001	5.097	0.050
Medication for cognition	Yes No *	−0.850 (1.121)	−3.086	1.387	0.451

Dependent variable: TYM—Total Score. B: Beta; SE: Standard error; *p*: *p*-value; CI: Confidence interval; R^2^ = 0.363; * reference category.

## Data Availability

The data presented in this study are available on request from the corresponding author. The data are not publicly available due to privacy.

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
