# Peer review of "Differences in Cognitive Function in Women and Men with Diabetic Peripheral Neuropathy with or without Pain"

_ijerph, 2022, doi:10.3390/ijerph192417102_

Round 1

Reviewer 1 Report

In this paper, the authors analyzed the differences in cognitive function between women and men with type-2 diabetes mellitus and diabetic peripheral neuropathy with and without diabetic neuropathic pain, and investigated the factors associated with cognitive function in different genders. The topic is of interest to readers. I have some comments listed below.

1.       Title: the title is confusing. Please try to avoid using more than one “and” in one sentence.

2.       Introduction: The first time you use an abbreviation, it's important to spell out the full term and put the abbreviation in parentheses. Then, you can use just the abbreviation in subsequent references after that. Please spell out the whole terms of DNP in line 73 as suggested.

3.       Method:

1)      Selection process: did the authors use a second method to confirm the existence of diabetic neuropathy besides monofilament test?

2)      For the diagnosis of diabetic neuropathy, how did the authors minimize the observer bias in the patients developed with foot ulcers?

4.       Results:

1)      Line 169: 83 were men. “Were” was left out.

2)      Line 170: there is no significant difference between female and male regarding the age. Please correct it.

3)      Table 1: is the p value for “education level” 0.426? Please correct the comma to dot.

4)      Table 1: there is a significantly higher rate in women compared to men regarding medication usage (affecting cognition). Should the patients using cognition affecting medications be excluded? Because this will be a great interference factor for the comparison between groups of regarding the TYM. Could the authors please explain?

5)      Table 3: the authors should also add the comparison of the clinical data (e.g. medications and complications) between the groups of male and female, with or without DNP, which will interfere the interpretation of the results.

6)      Table 1 and 3: there is no need to show both values (Yes or No) for qualitative variables in the table.

English editing needs improvement. There are concerns regarding patients’ recruiting and exclusion, which will interfere interpretation of the results.

Author Response

Reviewer 1

Comments and Suggestions for Authors

In this paper, the authors analyzed the differences in cognitive function between women and men with type-2 diabetes mellitus and diabetic peripheral neuropathy with and without diabetic neuropathic pain, and investigated the factors associated with cognitive function in different genders. The topic is of interest to readers. I have some comments listed below.

  1. Title: the title is confusing. Please try to avoid using more than one “and” in one sentence.

Response:

According to the recommendations, we have changed the title to read as follows:

“Differences in cognitive function in women and men with diabetic peripheral neuropathy with or without pain”

  1. Introduction: The first time you use an abbreviation, it's important to spell out the full term and put the abbreviation in parentheses. Then, you can use just the abbreviation in subsequent references after that. Please spell out the whole terms of DNP in line 73 as suggested.

Response:

Thank you for the suggestion. Diabetic neuropathic pain (DNP) has been added to the text. Line 81.

  1. Method:

1)      Selection process: did the authors use a second method to confirm the existence of diabetic neuropathy besides monofilament test?

Response:

The diagnosis was based on the patient’s medical records. As the patients had long-term diabetes, the disease was well known and detailed information was available in their clinical history. Furthermore, a thorough medical examination was performed of the patients’ feet using a validated instrument that has been shown to be effective at detecting diabetic neuropathy, as shown by the following reference that has been included in the article:

Zhang, Q.; Yi, N.; Liu, S.; Zheng, H.; Qiao, X.; Xiong, Q.; Liu, X.; Zhang, S.; Wen, J.; Ye, H.; et al. Easier Operation and Similar Power of 10 g Monofilament Test for Screening Diabetic Peripheral Neuropathy. Journal of International Medical Research 2018, 46, 3278–3284, doi:10.1177/0300060518775244.

In order to clarify this, the following sentence has been included in the Methodology section about the monofilament test, along with the reference about its qualities and use: “This procedure involves examining 10 reference points on each foot, 8 plantar and 2 dorsal. Diabetic neuropathy was confirmed if the patient did not detect the monofilament in three or more points on one foot”. Line 111-114.

2)      For the diagnosis of diabetic neuropathy, how did the authors minimize the observer bias in the patients developed with foot ulcers?

Response:

Thanks for your comment. We agree with the reviewer that there could be observer bias in the study. However, as mentioned in the response to the previous comment, to minimize any possible bias the evaluation of the patients was performed using a clinical examination, their medical records and the monofilament test. Likewise, most of the patients were long-term diabetes patients with well-known and detailed medical records.

  1. Results:

1)      Line 169: 83 were men. “Were” was left out.

Response:

In accordance with the suggestions made by the reviewer, the verb has been changed and the whole manuscript has been sent to an experienced translator of scientific English for proofreading. Line 183.

2)      Line 170: there is no significant difference between female and male regarding the age. Please correct it.

Response:

Thank you for your comment. The text only gives the mean age in each sex, with no intention of performing any kind of comparison.

3)      Table 1: is the p value for “education level” 0.426? Please correct the comma to dot.

Response:

Thank you very much. The comma has been changed to a dot.

4)      Table 1: there is a significantly higher rate in women compared to men regarding medication usage (affecting cognition). Should the patients using cognition affecting medications be excluded? Because this will be a great interference factor for the comparison between groups of regarding the TYM. Could the authors please explain?

Response:

In response to this comment, we must clarify that patients taking medication with an effect on cognitive function were not excluded from the sample since the variable “medication that affected cognition” was an adjustment variable in the multiple linear regression models, as explained in the Results section when describing both models. Excluding the patients fulfilling this characteristic would have considerably reduced the number of subjects included in the models, thus limiting the results obtained.

5)      Table 3: the authors should also add the comparison of the clinical data (e.g. medications and complications) between the groups of male and female, with or without DNP, which will interfere the interpretation of the results.

Response:

As the reviewer will have seen, Table 3 specifically includes the differences between men and women in the TYM and its dimensions. We considered that as cognitive function measured with the TYM is the dependent variable and the initial aim of our study, the inclusion of differences between men and women with and without neuropathic pain would increase the length of the manuscript without adding information of great relevance. However, in response to the reviewer’s comment, we have added the table suggested to the Supplementary Material.

6)      Table 1 and 3: there is no need to show both values (Yes or No) for qualitative variables in the table.

Response:

In response to your comment, we have removed the “no” from Table 1.

English editing needs improvement. There are concerns regarding patients’ recruiting and exclusion, which will interfere interpretation of the results.

Response:

The manuscript has been sent to a translator with experience in scientific English to improve the language used.

Reviewer 2 Report

 The aim of this study is interesting and helpful, and the results may facilitate the implementation of widescale mental health promotion strategies.

This research is well done and the methodology and results are well structured and presented, but the abstract is not very coherently written.

In this cross-sectional study, the authors evaluated the associations between cognitive impairment and diabetic peripheral neuropathy in persons with type 2 diabetes with or without diabetic neuropathic pain. The Instruments and variables are presented in a comprehensive manner.

For enhancing the information regarding the different results reported for the difference of QoL scale scores in type 2 diabetes patients according to sex/gender, I invite the authors to read and consider another paper such as 10.3390/ijerph18063249 (see the results for SF-36 scale, physical component score).

Please replace the correct form for HbA1c throughout the text.

Author Response

Reviewer 2

Comments and Suggestions for Authors

 The aim of this study is interesting and helpful, and the results may facilitate the implementation of widescale mental health promotion strategies.

  1. This research is well done and the methodology and results are well structured and presented, but the abstract is not very coherently written.

Response:

Thank you for the recommendation. The Abstract has been changed to make it more coherent.

In this cross-sectional study, the authors evaluated the associations between cognitive impairment and diabetic peripheral neuropathy in persons with type 2 diabetes with or without diabetic neuropathic pain. The Instruments and variables are presented in a comprehensive manner.

Response:

We would like to thank the reviewer for their comments.

  1. For enhancing the information regarding the different results reported for the difference of QoL scale scores in type 2 diabetes patients according to sex/gender, I invite the authors to read and consider another paper such as 3390/ijerph18063249(see the results for SF-36 scale, physical component score).

Response:

As suggested by the reviewer, in order to enhance the information provided, reference 7 has been added to the Introduction and the sentence:

“Likewise, men with DMT2 report having a better physical quality of life than women” Line 58 and 59.

  1. Please replace the correct form for HbA1cthroughout the text.

Response:

HbA1c has been replaced by HbA1c throughout the text.

Round 2

Reviewer 1 Report

The authors have addressed my concerns. I have no further queries.